# Modifications in Ultrastructural Characteristics and Redox Status of Plants under Environmental Stress: A Review

**DOI:** 10.3390/plants12081666

**Published:** 2023-04-16

**Authors:** Hana Ďúranová, Veronika Šimora, Ľuba Ďurišová, Lucia Olexiková, Marek Kovár, Miroslava Požgajová

**Affiliations:** 1AgroBioTech Research Centre, Slovak University of Agriculture, Trieda Andreja Hlinku 2, 949 76 Nitra, Slovakia; veronika.simora@uniag.sk; 2Institute of Plant and Environmental Sciences, Faculty of Agrobiology and Food Resources, Slovak University of Agriculture in Nitra, Trieda Andreja Hlinku 2, 949 76 Nitra, Slovakia; luba.durisova@uniag.sk (Ľ.Ď.); marek.kovar@uniag.sk (M.K.); 3Agricultural and Food Centre (NPPC), Research Institute for Animal Production Nitra, Hlohovecká 2, 951 41 Lužianky, Slovakia; luciaolexik@gmail.com

**Keywords:** environmental stress, abiotic stimuli, reactive oxygen species, antioxidant defense system, subcellular organelles, root tip, statocytes

## Abstract

The rate of global environmental change is unprecedented, with climate change causing an increase in the oscillation and intensification of various abiotic stress factors that have negative impacts on crop production. This issue has become an alarming global concern, especially for countries already facing the threat of food insecurity. Abiotic stressors, such as drought, salinity, extreme temperatures, and metal (nanoparticle) toxicities, are recognized as major constraints in agriculture, and are closely associated with the crop yield penalty and losses in food supply. In order to combat abiotic stress, it is important to understand how plant organs adapt to changing conditions, as this can help produce more stress-resistant or stress-tolerant plants. The investigation of plant tissue ultrastructure and subcellular components can provide valuable insights into plant responses to abiotic stress-related stimuli. In particular, the columella cells (statocytes) of the root cap exhibit a unique architecture that is easily recognizable under a transmission electron microscope, making them a useful experimental model for ultrastructural observations. In combination with the assessment of plant oxidative/antioxidative status, both approaches can shed more light on the cellular and molecular mechanisms involved in plant adaptation to environmental cues. This review summarizes life-threatening factors of the changing environment that lead to stress-related damage to plants, with an emphasis on their subcellular components. Additionally, selected plant responses to such conditions in the context of their ability to adapt and survive in a challenging environment are also described.

## 1. Introduction

Drastic and rapid global climate change, such as global warming and elevated atmospheric carbon dioxide (CO_2_), as well as other impacts of anthropogenic activities, such as desertification, soil salinization, and nutrient imbalances (mineral toxicities/deficiencies) have induced severe modifications in the agricultural land environment. These interventions have principally led to a radical shift in crop production worldwide [1,2] and in the distribution of plant species among ecological niches [3]. Combined with rapid global population growth, the agricultural sector is hence increasingly challenged to meet global food security [4]. In order to face this global issue, the scientific community is now paying close attention to concerns about environmental stability and crops’ adaptability to changing conditions [5].

In general, the adverse action of non-living (abiotic) factors on living beings in a specific environmental setting is known as abiotic stress [2,6]. Drought, flood, high soil salinity/sodicity, extreme temperatures, oxidative stress, heavy metal toxicity, and ultraviolet (UV) radiation are the most common harmful environmental stressors that strongly affect the growth, development, and survival of crops, resulting in their reduced distribution, productivity, and sustainability [7,8,9]. These abiotic stressors are often interrelated, so they can act not only individually but also in diverse combinations [10,11,12]. According to Dos Reis et al. [1], more than 90% of arable lands worldwide are estimated to be subjected to one or more of the substantial components of the environment, with a 50–70% yield decline in staple food crops [13,14,15].

Essentially, plants respond to environmental factors at multiple levels (such as anatomical, morphological, cellular, molecular, and whole-plant physiological levels), and these responses can differ depending on the species, genotype, age, and developmental stage of the plant, as well as on the duration and severity of the event [16]. Furthermore, plants are equipped with more generalized cellular, morphological, and physiological defenses, and adaptive tactics to counter abiotic stress factors. These include the cuticle (the outermost shield), unsaturated fatty acids (UFAs; membrane modulator and oxylipin precursors), reactive species (RS) scavengers (governing RS homeostasis), molecular chaperones (stabilizing proteins), subcellular structures (e.g., the membrane) or compatible solutes (acting as more than osmoprotectants) inside the cells [9]. Thus, it is essential to comprehend the mechanisms by which plants perceive stress signals and adapt to unfavorable abiotic environmental conditions at the cellular and molecular levels. This understanding is a vital prerequisite for the development of abiotic stress-resistant (tolerant) plants, which is crucial for meeting the demands of population growth and ensuring global crop availability [1,17]. 

The present review provides a comprehensive overview of key abiotic factors affecting plants and their responses to stress, with a particular focus on the impact of oxidative changes and alterations in cellular ultrastructure. Moreover, the sophisticated adaptation machinery and complex regulatory networks that ensure the adaptation/tolerance strategies of plants are briefly described. 

## 2. Abiotic Environmental Stress Factors

In plants, the most important abiotic factors participating in their major physiological processes comprise light, CO_2_, water, temperature, nutrients, and salinity. Indeed, their optimal availability is necessary to fulfill the plant’s basic requirements; however, when the dosage of any of these factors is under or above optimum, the agent can become and act as a stress factor or stressor [2,18]. As plants are sessile organisms that are incapable of movement, they are inevitably exposed to a spectrum of abiotic stresses, such as drought, salinity, and extreme temperatures. Consequently, they must possess the ability to withstand such stresses and rapidly acclimate to new local conditions [17]. Moreover, as mentioned already, the abiotic stresses are interconnected with each other [11], thus generating complex deleterious conditions that can destabilize agricultural systems [12,19].

### 2.1. Drought (Water Stress)

Drought is commonly induced by an absence of water as a consequence of irregular rainfall or insufficient irrigation; however, it can also be connected with salinity and the physical properties of soil, and high air or soil temperature [20]. Since water is a critical input to plant productivity, its deficiency is one of the principal environmental constraints on agricultural production. Indeed, it is true that a drought-induced decline in crop yield can, depending on the severity and stress duration (temporary or permanent), exceed all losses caused by any other abiotic stress factors [21,22].

As a multidimensional stressor, drought induces changes in the morphological, physiological, biochemical, and molecular traits of vascular plants. Morphologically, plants experiencing water stress are characterized by reduced total plant height [23], number and size of leaves, and stem extension [20,22,24]. Leaf rolling and root length increment are other growth pattern of plants under such conditions [25]. The alterations following disturbed plant–water relationships and reduced water-use efficiency [20] are mainly linked to the decrease in leaf water potential and turgor pressure, stomatal closure [22], and changes in stomatal conductance and distribution [25], effectively causing the elimination of water loss via transpiration [3]; the overproduction of reactive oxygen species (ROS); modifications to stress signaling pathways [26]; and alterations in photosynthesis, chlorophyll synthesis, ion uptake and translocation, and nutrient and carbohydrate metabolism [27]. In addition, plants exposed to drought activate various strategies and response mechanisms, such as enhancing their root system architecture, altering the dynamics of their root-to-shoot ratio, accumulating compatible solutes (such as proline), upregulating the expression of drought-resistant genes, synthesizing hormones (e.g., stress-induced phytohormones) and osmotic regulatory substances, stimulating their antioxidant defense system, and delaying senescence [23,25,28].

### 2.2. Salinity (Salt Stress)

High salt concentration in the soil is another worldwide problem that severely restricts crop productivity and sustainability, and limits the sustainable development of modern agriculture [29]. This abiotic stressor is especially a result of improper irrigation, a lack of drainage, and the excessive accumulation of soluble salts [30]. According to estimates mentioned by Hernández [31], salinity affects approximately 800 million hectares of land worldwide. Additionally, over 6% of agricultural land globally is at risk of disappearing [30].

On a global scale, sodium (Na^+^) and chloride (Cl^−^) ions have been recognized as the main contributors to salt toxicity, influencing up to 50% of irrigated soils [32]. In addition to the specific toxicities, plant growth retardation in saline soils can also be induced by osmotic stress and reduced uptake of essential macro- and micronutrients [33]. Excess Na^+^ influx and potassium (K^+^) efflux can also disrupt the ionic homeostasis of plants, which is a frequently occurring phenomenon [34]. In addition, salinity exposure can lead to various other effects, such as hormonal imbalances; the initiation of oxidative stress; alterations in photosynthesis, transpiration, and stomatal conductance; and alterations in endogenous phytohormonal functions, essential metabolic pathways, and gene expression patterns [35,36,37]. Due to stress caused by increased salinity, the enhanced formation of ROS in plants leads to increased activity of the antioxidant enzymes superoxide dismutase (SOD), catalase (CAT), peroxidase (POX), glutathione peroxidase (GPX), or ascorbate peroxidase (APX), and the elevation of non-enzymatic antioxidants including ascorbic acid and glutathione (GSH). Moreover, stress-induced nitric oxide (NO) formation potentiates the expression of genes that encode antioxidant enzymes and redox-related molecules [35,36]. Salt stress can also have detrimental effects on seed germination and seedling establishment by reducing the levels of seed germination stimulants such as gibberellic acid, increasing the levels of abscisic acid (ABA), altering membrane permeability, and affecting water behavior in seeds through induced osmotic and oxidative stresses, as well as ion toxicity [38]. In fact, Oyiga et al. [39] found that salt stress led to a decrease in the germination vigor, seedling shoot dry matter, and seed grain yield of wheat genotypes by 33%, 51%, and 82%, respectively.

To cope with salinity, plants utilize diverse pleiotropic mitigating strategies. These include accumulating solutes in their roots to reduce internal water potential, altering their proline metabolism in response to changes in hormonal signaling, using cellular signaling to expel Na^+^ ions, utilizing high-affinity K transporters (HKTs) to counteract Na toxicity, and activating and promoting different genes to maintain ion homeostasis [37]. Additionally, phytohormones play a crucial role in regulating plant growth adaptation to salinity [40]. As an adaptive response, the roots are also able to modulate metabolism, gene expression, and protein activity, which ultimately results in modifications to their cell wall composition, transport processes, cell size and shape, and overall structure [41].

### 2.3. Extreme Temperature (Cold/Heat Stress)

Extreme temperatures (high or low) are among the primary abiotic factors that contemporaneously influence the action of other ones, and they also have an undesirable impact on plant development and biomass production [2]. Heat stress, in particular, is frequently compounded by additional abiotic stresses such as drought and salinity. However, it also has an independent mode of action on the physiology and metabolism of plant cells [10]. In this context, it has been shown that an increase in the seasonal average temperature by 1 °C can lead to a decrease in the grain yield of cereals by 4.1–10.0% [42].

Across all plant species, the most sensitive phenological stage to temperature extremes is pollination [43]. During the development stage, these extremes can greatly reduce plant pollination rates and, thus, affect plant production. For instance, warm temperatures have been found to reduce the quantities and alter the compositions of floral resources. This results in reduced nectar volume and total quantity of nectar sugars and pollen weight per flower, an increase in the nectar total amino acid concentration and essential amino acid percentage, and a rise in pollen polypeptide concentration [44]. In addition, warm temperatures can decrease flower attractiveness and bumblebee foraging in the entomophilous species *Borago officinalis* [45]. Furthermore, heat stress eliminated the function of tapetal cells, caused anther dysplasia, and reduced the percentage of seed germination and photosynthetic efficiency [11]. Improper development; alterations in the growth period and distribution of crop plants; changes in dry matter partitioning; restrictions in the cell division process; the generation of oxidative stress, together with membrane and protein damage; the disruption of biomolecule synthesis [42,46]; reduced ribulose-1,5-bisphosphate carboxylase/oxygenase (Rubisco) activity; and the inactivation of photosystem II (PSII) electron acceptor and donor sides, and Calvin cycle enzymes [19], were also observed in plants that encountered high temperatures. To address this issue, the establishment of a molecular ‘thermomemory’ after moderate heat stress exposure, which allows plants to withstand a potentially more extreme heat stress event, has been proposed as a potentially powerful strategy for enhancing plant survival and reproduction under heat stress [47].

On the other hand, low temperature, as the main determinant of chilling and freezing stresses, can induce flower and/or leaf injury and damage with subsequent marked yield losses [14]. Commonly, cold stress affects plant cellular functions at every level. In the cell environment, it leads to the formation of ice crystals, which consequently reduces plant cellular metabolism and can also result in cell death because of dehydration and electrolyte leakage [46]. To cope with cold stress, plants use small signaling molecules for cellular signal transduction, such as calcium (Ca), ROS, NO, hydrogen sulfide, cyclic guanosine monophosphate (cGMP), protein kinase, phosphatidic acid, ABA, and sphingolipids, which activate downstream signaling cascades. These signaling molecules are involved in various physiological processes, including the induction of gene expression, the activation of hormone signaling, the upregulation of antioxidant enzyme activities, the accumulation of osmoprotectants, a reduction in malondialdehyde (MDA), and an improvement in photosynthesis [11,48].

### 2.4. Metal and Nanoparticle Toxicity Stress

The soil–plant system is heavily impacted by toxic metals and nanoparticles, with chemical fertilizers, rapid industrialization, and sewage wastewater irrigation in agriculture being the main anthropogenic causes, as noted by Gull et al. [11]. Among them, titanium dioxide (TiO_2_), silver (Ag), zinc oxide (ZnO), cerium dioxide (CeO_2_), copper (Cu), copper oxide (CuO), aluminum (Al), nickel (Ni), and iron (Fe) are the most commonly used in industries and, thus, they are the most investigated for their impacts on diverse plant species [49]. 

Heavy metal stress in plants is commonly associated with the excessive generation of ROS and the induction of oxidative stress, which has the ability to initiate oxidative damage to several vital cellular biomolecules, including DNA and proteins [50]. As a result, diverse morphological, physiological, and metabolic anomalies may occur, such as lipid peroxidation, protein degradation, and chlorosis of the shoot [51]. From anatomical and morphological perspectives, heavy metal-induced toxicity in plants can be characterized by several modifications, such as a higher degree of root lignification and suberization [52], root cell walls modifications, changes in the development of apoplastic barriers (particularly Casparian bands and suberin lamellae in the exo- and endo-dermis) and vascular elements (e.g., enhanced lignification of xylem elements, alterations in size and transport capacity), a reduction in root and stem diameter, decreased leaf thickness, structural alterations in mesophyll tissues, increased intracellular spaces in cortical tissues, enhanced pit parenchyma, and modifications in stomatal frequency [53]. These modifications in the structural and physiological integrity of leaves are characterized by an overall decrease in the number and size of leaves, their enhanced rolling and abscission, decreased chlorophyll content, chloroplast malformation, changes in stomatal and guard cell size, inhibition of the electron transport chain, reduced CO_2_ fixation [52], and reduced rates of photosynthesis and respiration [54]. Not only can heavy metals seriously affect plant growth and development, but so can metal oxide nanoparticles. They have been shown to cause oxidative stress (leading to the disruption of cellular metabolism), and to reduce seed germination, root/shoot growth, biomass production, and the biochemical or physiological activities of plants [55].

Plants have developed and adopted a plethora of adaptive strategies to counteract the toxicity of metals and/or nanoparticles. These strategies include the formation of metal complexes through chelation with organic molecules, such as phytochelatins and metallothioneins, at both the intra- and inter-cellular levels. Additionally, plants activate antioxidative enzymes and other non-enzymatic mechanisms to prevent or repair various oxidative stress-induced secondary defects [51,52,54,55,56,57]. The enhanced production of phytohormones, particularly auxin [58,59], and non-enzymatically synthesized compounds (e.g., proline) [51] also enhances intracellular antioxidant enzyme capacity for metal detoxification. Other mechanisms that contribute to plant tolerance to heavy metal stress, mediated by phytohormones, include the improvement of osmoregulation, photosynthetic and gaseous exchange traits, GSH production, and metal transporter induction [58]. 

## 3. Some Selected Plant Defense Responses and Their Activated Mechanisms against Abiotic Constraints, with Special Emphasis on Oxidative/Antioxidative Status and Cell Ultrastructure 

### 3.1. Protein Quality Control Systems—Protein Folding Stability and Dynamics 

For the survival of plant cells undergoing abiotic stress, it is crucial to maintain the functional native (correctly folded) conformation of their proteins and prevent the aggregation of non-native proteins, which can cause metabolic disruptions [60]. In fact, protein dysfunctions based on protein unfolding, misfolding, and aggregation are common causes of stress conditions in exposed plant cells [61]. However, to maintain protein homeostasis, plants have evolved an extensive protein quality control system, which includes heat shock proteins (HSPs), the unfolded protein response (UPR) and autophagy, and the ubiquitin–proteasome system (UPS) [62].

Heat shock proteins, functioning as molecular chaperones, are the critical components responsible for protein folding, assembly, translocation, and degradation during cellular processes under normal and stressful conditions. These proteins play an essential role in maintaining cellular homeostasis and conferring plant tolerance by stabilizing proteins and membranes, and promoting protein refolding to restore their normal conformation during abiotic stress [63,64]. Furthermore, HSPs have been found to enhance membrane stability and detoxify ROS by mediating the regulation of antioxidant enzymes [60]. The five conserved HSP classes, including HSP100/Clp, HSP90, HSP70/DnaK, HSP60/Chaperonin, and small HSP (sHSP), have been identified based on molecular weight. HSP70, the most conserved class across different species, consists of an N-terminal ATPase domain and a C-terminal substrate-binding domain [9]. Its up-regulated accumulation has been reported in *Brachypodium* seedling roots under 15% polyethylene glycol-induced osmotic [65] and hydrogen peroxide (H_2_O_2_) stress [66]. 

The UPR is a highly conserved stress response that is triggered by the accumulation of misfolded proteins in the lumen of the endoplasmic reticulum (ER) [67]. This subcellular organelle is primarily recognized as a major site for protein folding, and it plays a pivotal role in ensuring the proper folding and maturation of newly secreted proteins [68]. Aberrations in this process, known as ER stress, can disrupt protein homeostasis. To restore protein homeostasis, the highly specific cell-signaling system UPR, with inositol-requiring enzyme 1 (IRE1) as a signal activator, readjusts the folding capacity of the organelle [69]. In addition to adaptation to ER stress, the system is essential for root growth and development, and also protects male gametophyte development from heat stress [67]. In plants, there are two branches of the UPR signaling transduction pathway: one involving the proteolytic processing of two ER membrane-associated basic leucine zipper domain (bZIP17/28) transcription factors, and another involving a dual protein kinase (RNA-splicing factor IRE1) and its target RNA (bZIP60) [70,71]. Depending on stress conditions, the UPR can be classified into two separate phases: (i) the adaptive UPR, i.e., the activation of autophagy (cell survival response) via signaling from IRE1 under mild or short-term stress circumstances, and (ii) the apoptotic UPR (leading to cell death) under severe or chronic stress conditions [70]. Autophagy ensures the degradation and recycling of damaged proteins, protein aggregates, and whole organelles as a functioning mechanism of plant abiotic stress management [72].

The UPS regulates the abundance of numerous enzymes, as well as structural and regulatory proteins, thereby maintaining many cellular functions and processes. During environmental stress conditions, it plays a key role in the modification of protein load in the cell via degradation [73]. The system involves the attachment of a chain of ubiquitin molecules to a targeted protein and its degradation by the multi-proteolytic 26S proteasome, also recycling the ubiquitin molecules [74]. The sequential action of three types of enzyme—E1 (ubiquitin activating enzyme; UBA), E2 (ubiquitin conjugating enzyme; UBC), and E3 (ubiquitin ligase)—is involved in the process of ubiquitination [75]. The process has a crucial role not only in the abiotic stress responses of plants, but also in the cell cycle, intracellular trafficking [76], immunity, and hormonal signaling [73], via interference with key components of the pathways. Additionally, it participates in the uptake, transport, and homeostasis of nutrients (such as Fe, phosphorus, and nitrogen) [77].

### 3.2. Osmoregulation and Compatible Solutes

Under abiotic stress, osmoregulation is a common plant defensive strategy that relies on the synthesis and accumulation of osmoprotectants or compatible solutes, acting as osmolytes to protect the cellular machinery. These low-molecular-weight compounds include polyamines (an ammonium compound group), glycine betaine (a quaternary amine), amino acids such as proline, asparagine, and serine, γ-amino-N-butyric acid (GABA), soluble proteins, sugars such as trehalose, and sugar alcohols such as inositol and mannitol [78,79,80]. These nontoxic osmolytes stabilize osmotic differences between the surroundings of the cell and the cytosol, balance the osmotic potential of Na^+^ and Cl^−^ accumulated in the vacuole, protect cellular structures and enzymes, and also act as metabolic signals and scavengers of ROS produced under stressful conditions [80,81]. In the plastid stroma, accumulated soluble osmolytes not only reduce stromal water potential (preventing tissue dehydration), but also function as cryoprotectants that protect proteins and membranes from inactivation and lipid peroxidation, respectively. They also serve as an energy reservoir necessary for the energy-consuming processes of plant adaptation [82]. 

In addition to this, compatible solutes are commonly characterized by a high level of solubility in the cellular environment, keeping the main physiological functions of the cells, and by the absence of enzyme activity inhibition (even at high concentrations). Moreover, their biosynthesis is controlled by environmental cues, and it can be realized at all developmental stages [78,83]. Osmolytes accumulation can be further modulated by phytohormones, including ABA, brassinosteroids, cytokinins, ethylene, jasmonates, and salicylic acid [81].

### 3.3. Unsaturated Fatty Acids as General Defenders

As one of the constitutive and inducible tactics to changing the environment, UFAs have multiple roles in plants. They serve as ingredients and modulators of cellular membranes, a carbon and energy reservoir, extracellular barrier constituent storage, precursors of various bioactive molecules (e.g., jasmonates and nitroalkenes), and regulators of stress signaling [84]. Among them, the C18 species, especially oleic (18:1), linoleic (18:2), and α-linolenic (18:3) acids, are the major UFAs and play a crucial role in the plant defense system [85].

To cope with stress conditions, changes at the level of UFAs (via the regulation of fatty acid desaturase activity) induce modifications in membrane fluidity, which are of profound importance to maintain an environment suitable for the function of critical integral proteins (e.g., photosynthetic machinery) [86]. In addition to the degree of fatty acid desaturation, the acyl chain length and positional distribution of UFAs on the glycerol backbone are also fundamental keys to membrane organization and dynamics [87]. For instance, it has been found that elevated levels of membrane UFAs and/or phosphatidylglycerol (an integral component of photosynthetic membranes) can alleviate PSII and/or photosystem I (PSI) photoinhibition in plants under chilling and/or salt stress conditions [88,89,90]. Furthermore, the unsaturation of fatty acids has also been linked to the repair of the PSII complex following damage due to strong light action [91]. The results of Allakhverdiev et al. [92] suggest that the protection of PSII and PSI against NaCl-induced inactivation, and acceleration of the recovery of both photosystem activities, might be associated with the unsaturation of fatty acids in membranes. This could stimulate the synthesis of Na^+^/H^+^ antiporter(s) and/or H^+^-ATPase(s) in the membrane, leading to a decline in Na^+^ ion concentrations in the cytosol.

As a raw material, plant C18 UFAs are also utilized to produce a plethora of aliphatic compounds, including triacylglycerols (TAGs), cutin/suberin, jasmonates, and nitroalkenes (NO_2_-FAs) [84]. In plants, TAGs, as a major cellular carbon and energy reservoir, are synthesized in minor amounts in the ER and accumulated in lipid droplets (LDs) in the cytosol or in chloroplast stroma in plastoglobules [93]. They are commonly involved in cell division and expansion, stomatal opening, membrane lipid remodeling, flower development, and pollination [94]. Typically, TAGs represent < 1% of leaf glycerolipids; however, their intensive synthesis and accumulation have been observed under stress conditions [95]. Indeed, the accumulation of large amounts of TAGs in LDs in stressed photosynthetic cells is believed to serve as a storage source for fatty acids, required to satisfy the changing cellular demands for energy, metabolism, and growth [96], and for further membrane regeneration after stress cessation. In TAG biosynthesis, the up-regulation of several genes, such as diacylglycerol O-acyltransferase (DGAT2), which is the most important one, is involved [97]. A study performed by Chen et al. [98] revealed up-regulation of GmDGAT2D in soybean seeds subjected to heat and cold stress. Additionally, drought stress [97,99,100], metal toxicity stress [101], and nutrition deprivation [102] have been shown to induce the stimulated accumulation of TAGs in various plants. In addition to this, TAGs under stress conditions are also believed to function as a transit pool to sequester the over accumulation of toxic polyunsaturated free fatty acids and other lipid intermediates, thus preventing cellular damage [97,103]. 

### 3.4. Oxidative/Antioxidative Stress Concept

Generally, stress caused by challenging conditions in the environment activates plant immune responses that are associated with the rapid production of ROS and reactive nitrogen species (RNS) that dramatically alter cellular redox homeostasis [104]. Redox chemistry is one of the key features of living organisms. Oxidized substrates are reduced to promote the synthesis of functional molecules, and reduced substrates are oxidized to ensure energy supply [105]. Abiotic stress is tightly associated with fluctuations in cellular redox, as cells must fight against the uncontrolled oxidation of essential cellular components [106]. Redox homeostasis, thus, depends on the simultaneous collaboration of the complex network of redox-active compounds, such as ascorbate (AsA), GSH, NAD(P)H, and proteins from the thioredoxin (TRX) superfamily, to allow for normal physiological responses and adaptation to stress conditions [107]. All components of the “redox-hub” are ubiquitously present in the cytosol and subcellular organelles, including chloroplasts, mitochondria, and peroxisomes. Overall, the concept of cellular redox status represents a qualitative, hypothetical state combining the oxidation/reduction conditions and distinct redox-active metabolites, which regulate antioxidant–ROS balance (Figure 1) in stress biology [108].

#### 3.4.1. Reactive Oxygen Species 

Under physiological conditions, more than 90% of oxygen molecules consumed by living plant cells are used in the electron transport chain as terminal acceptors of electrons. The remaining oxygen enters partial one-electron reduction via the consequent addition of electrons, resulting in the formation of oxygen products, known as ROS. The key producers of ROS under normal or stress conditions in plants are plasma membrane NADPH oxidases, e.g., the respiratory burst oxidase homologues (RBOHs). In higher plants, ROS such as superoxide radical (O_2_•^−^), singlet oxygen (^1^O_2_), H_2_O_2_, perhydroxyl radical O_2_H•, or hydroxyl radical (OH•) were found to regulate various physiological processes including differentiation, development, stress signaling, redox levels, long-distance signaling, immune response, cell death, etc. [109,110,111].

Progressive molecular oxygen reduction occurs as follows [112]:O_2_ + e^−^+ H^+^→ HO_2_•
HO_2_• → H^+^ + O_2_•^−^
O_2_•^−^ + 2H^+^ e^−^→ H_2_O_2_
H_2_O_2_ + e^−^→ HO^−^ + HO•
HO• + H^+^ + e^−^→ H_2_O

As toxic byproducts of aerobic metabolism, ROS are formed in any cellular compartment that comprises molecules with a sufficiently high redox potential to excite or donate an electron to atmospheric oxygen; however, the primary ROS generation organelles consist of chloroplasts, mitochondria, and peroxisomes. The most prominent electron donor sources that result in ROS generation are depicted in Table 1. The conversion of oxygen intermediates to highly reactive forms involves transition metal ions, particularly Fe, which reacts with H_2_O_2_ (Fenton reaction) to generate OH• that causes damage to most organic molecules [113]. Due to the high reactivity of ROS or free radicals, their levels are kept under control to prevent unintended cellular oxidation. Thus, formed ROS are removed or detoxified by pathways of enzymatic defense systems or compounds with antioxidant properties to keep ROS at low levels, and to control ROS signaling reactions [114,115]. Perturbation of the equilibrium between the generation and sequestration of ROS due to adverse environmental factors often results in a rapid increase in intracellular levels of ROS [116]. Excess ROS is a common feature of abiotic and biotic stress, contributing to the enhanced oxidative damage of cellular structures [117,118]. Several core stress signaling pathways participate in salt resistance, including ROS homeostasis maintenance [119,120]. High light-derived accumulation of ROS leads to photodamage to chloroplasts and a decrease in their size, and changes in their ultrastructure can occur [121,122]. Drought and cold stresses result in stomata closure that initiate signaling through components including ROS, NO, cytosolic pH, or free Ca^2+^ [123]. Other types of abiotic stress, such as heavy metal exposure [51,124], extreme temperatures [125], or nutritional deprivation [126], lead to ROS overproduction, causing damage to proteins, lipids, carbohydrates, and DNA, which ultimately results in oxidative stress [127,128,129].

#### 3.4.2. Oxidative Stress

In the context of living organisms, stress is unavoidable and natural as unstressed conditions are almost unattainable. Organisms are exposed to stress on an everyday basis and respond to it to maintain survival. Although there is no accurate definition of oxidative stress, generally, the imbalance between ROS generation and their elimination, with certain consequences for cell physiology, is referred to as oxidative stress [130]. Unlike animals, plants cannot respond to the changes in their environment by moving to a more suitable environment; hence, the primary response of a cell to oxidative stress is the activation of multiple defense, adaptive, and acclimation mechanisms that enable the cell to survive after a dramatic change in metabolic conditions [131,132]. Since the Earth’s surface underwent stepwise oxygenation between approximately 2.8 and 0.8 Ga, organisms were forced to exploit oxygen to permit life under aerobic conditions [133,134]. This led to the formation of more complex organisms and prompted biodiversity. However, high levels of ROS, including ^1^O_2_, OH•, O_2_^•−^, O_2_H•, or H_2_O_2_, determine various responses, resulting in the induction, reduction, or inhibition of growth, and alternatively, induce tolerance, acclimation, or defense against the stressor. In all cases, uncontrolled excess ROS can cause structural modification and functional alteration in nucleic acids, proteins, and lipids, ultimately leading to cell and organism death [132]. For plants, higher concentrations of ROS have injurious impacts on cellular molecules, organelles, and tissues of the shoots and roots. Hence, inevitable consequences of oxidative stress, such as impaired crop quality and yield, can occur [135]. Understanding stress and its outcomes allows for the optimal cultivation of plants. It has been recognized that the oxidative degradation of lipids results in MDA formation and yields to the formation of 4-hydroxynonenal and isoprostanes from UFAs. Oxidative stress-induced alterations in proteins, which lead to the loss of their activity, are a consequence of thiol oxidation, carbonylation, side-chain oxidation, fragmentation, unfolding, and misfolding. 8-hydroxydeoxyguanosine (8-OHdG) is an index of DNA damage caused by ROS [112]. Plants exposed to oxidative stress undoubtedly suffer from dramatic alterations at the transcriptome level that, together with the posttranscriptional, translational, and posttranslational regulations, shape the active proteome. As compared to less abiotic stress-tolerant species and cultivars, their counterparts have frequently shown enhanced gene expression and protein abundance of mitochondrially localized antioxidant defense genes [136,137]. Moreover, the genetic manipulation of dissipative mechanisms has revealed considerable success in the generation of stress-tolerant plants [138].

#### 3.4.3. Antioxidant Defense System

Protective mechanisms dedicated to maintaining physiological levels of ROS that combine the cooperation of preventive or repair systems and classes of antioxidants, including enzymatic and non-enzymatic, endogenous and exogenous, primary and secondary, hydrosoluble and liposoluble, natural or synthetic compounds, are all crucial for aerobic organisms [135]. The dissipative pathway to reduce ROS production in chloroplasts includes cyclic electron transport (CET) dedicated to equilibrating the photosynthetic requirements for energy and reducing power, chlororespiration through the plastid terminal oxidase (PTOX), and alternative electron sinks on the acceptor side of PSI. In mitochondria, ROS-reducing mechanisms involve the cyanide-insensitive alternative oxidase (AOX) and the uncoupling protein (UCP). All these pathways help to regulate excess energy and/or to reduce equivalents in the particular electron transport chains, and thereby limit excess ROS propagation. Moreover, photorespiration reduces chloroplast oxygen levels through the oxygenase activity of Rubisco, especially under the conditions of limited CO_2_ fixation [138].

Ferritins represent another example of ROS formation-reducing agents, that act through Fe sequestration. Free Fe can induce cellular devastation through the induction of Fenton-type reactions that lead to HO• production during the oxidation of Fe^2+^ by H_2_O_2_. Ferritins from various sources interact with Fe^2+^ to induce its oxidation in a reaction that is catalyzed by a ferroxidase center, which results in lower ROS accumulation and increased tolerance to various environmental stresses besides excess Fe [139,140].

As ROS overproduction results in oxidative stress, signaling through the transduction of signals from the central molecules in stress, i.e., mitogen-activated protein kinases (MAPKs), leads to the activation of stress-responsive gene expression. Thus, the activated MAPKs signaling through the induction of phosphorylation cascades initiates the detox antioxidant systems and adjusts ROS levels [141]. Non-enzymatic and/or enzymatic components of the antioxidant defense system scavenge or detoxify the excessive ROS, resulting in mitigation of the negative effects of oxidative stress. The most intensively studied enzymatic systems of the antioxidant defense network are SOD, CAT, APX, GPX, guaiacol peroxidase (POD), glutathione reductase (GR), monodehydroascorbate reductase (MDHAR), dehydroascorbate reductase (DHAR), glutathione S-transferase (GST), and TRX. The non-enzymatic defense compounds with high antioxidant properties include ascorbic acid, GSH, flavonoids, vitamin B6, carotenoids, tocopherols, and melatonin [127,142].

The most frequent targets of antioxidant enzymes under abiotic stresses are O_2_^•−^ and H_2_O_2_, of which O_2_^•−^ is converted to H_2_O_2_ by the metalloenzyme SOD in the chloroplasts, mitochondria, cytoplasm, apoplasts, and peroxisomes. Furthermore, catalases and peroxidases detoxify H_2_O_2_; however, as H_2_O_2_ at low concentrations plays an important role in signaling under stress, the detoxifying enzymes impose tight regulation on its cellular concentrations rather than its total clearing. As CAT has been shown to possess a lower affinity to H_2_O_2_ compared to POD (mM and µM range, respectively), CAT is involved in the mass scavenging of H_2_O_2_, whereas POD is believed to be involved in its fine tuning [143,144]. In addition, APX, with its high affinity for H_2_O_2_, acting in the chloroplasts, cytosol, mitochondria, peroxisomes, and apoplastic space, reduces H_2_O_2_ to H_2_O, utilizing AsA as a specific electron donor [145]. GPXs catalyze, in addition to the reduction of H_2_O_2_, reduction of cytotoxic hydroperoxides to alcohols, thereby detoxifying the products of lipid peroxidation formed by ROS [146]. The association with the activity modulation of enzymatic defense systems and changes in antioxidant gene expression in response to abiotic stress has been confirmed in many studies on various plants and stressors. Moreover, a number of transgenic plants, such as *Arabidopsis*, tomato, rice, tobacco, or maize, have been developed with modified expression of antioxidant enzymes, and have displayed alleviated tolerance to abiotic stresses including salinity, extreme temperatures, and drought [135].

Low-molecular-weight antioxidants, such as AsA, GSH, and tocopherols, can mitigate ROS-derived alterations in cells and cellular organelles. In the context of metabolic and environmental stimuli, AsA and GSH are the key players in a redox hub to tone the responses within the cellular signaling network [147,148]. The interplay between lipid-soluble α-tocopherol and water-soluble AsA and GSH antioxidants covers the conversion of lipid peroxyl radicals (LOO•), generated as a consequence of ROS-induced membrane lipid damage, to lipid hydroperoxides (LOOH) by α-tocopherol; meanwhile, the formed α-tocopheroxyl radical is reduced back to α-tocopherol by AsA. In addition, monodehydroascorbate (MDHA) and dehydroascorbate (DHA), the oxidized AsA forms, are produced during this process. Ascorbate restoration is achieved by MDHAR, which converts MDHA, and by DHAR, which reduces DHA back to AsA utilizing reduced GSH. Moreover, GR catalyzes the reduction of oxidized glutathione (GSSG) back to GSH. These interconnected mechanisms secure better tolerance against abiotic stresses, including salt, heavy metal, or osmotic stress in plants [149].

Cell survival, development, or adaptation to challenging conditions in their environment highly depends on mutual interactions in the surrounding environment and individual biochemical pathways that ensure responses to various stresses, while their imbalance results in oxidative damage and cell death (Figure 2).

### 3.5. Cell Ultrastructure (Subcellular Organelles) as a Reliable Abiotic Stress Marker and Its Adaptive Strategies

At its core, plant growth and development abnormalities are evident not only at the whole-plant level, but also at the cellular level, often manifesting through alterations in subcellular structures (Figure 3). Indeed, cellular organelles are crucial for stress tolerance, and perturbations in coordinated communication among cellular compartments are linked to responses of the cells to environmental stress circumstances [136]. The major subcellular components affected by abiotic stress include the cell wall, plasma membrane, nucleus, mitochondria, plastids, and endomembrane system - ER and Golgi apparatus (GA), whose ultrastructural modifications related to their functions can be visualized through transmission electron microscopy (TEM), as a result of negative stressor impacts or adaptive responses to changing homeostasis [151]. For instance, disorders linked to plasmolysis manifestation, the organelle moving, local changes in the cytoplasm density, transformations in the ER distribution, and the vacuolar compartment can be associated with the oxidative, osmotic, and toxic impacts of salt abiotic stress [152]. A summary of the impacts of numerous abiotic stress factors on plant qualitative ultrastructural characteristics is depicted in Table 2.

#### 3.5.1. Cell Wall and Plasma Membrane

In addition to its principal role in the control of plant growth and development, the cell wall participates in the acclimation processes of plants exposed to diverse abiotic stresses. One such stress is salt exposure, where the cell wall acts as a barrier to prevent the influx of Na^+^ and Cl^−^ ions into the protoplast, thus maintaining cell structure. A study conducted by Gao et al. [153] examined the effects of various concentrations of NaCl on leaf mesophyll cell ultrastructure in potato plantlets, revealing progressive modifications in the cell wall and plasma membrane with increasing salt concentration. Indeed, these alterations varied from twisted cell walls with remarkably crimped membranes (25 mM NaCl for 2 weeks) to cell plasmolysis accompanied by a reduction in mesophyll intercellular spaces (50 mM NaCl for 2 weeks), the presence of more vesicles in the vacuole (100 mM NaCl for 6 weeks), and damaged membrane structure characterized by severe membranous invaginations or ruptured cell walls with whole-cell disorganization (200 mM NaCl for 2 or 6 weeks). Partial plasmolysis with occasional detachment of the cytoplasmic membrane from the cell wall was also reported in *Arabidopsis* (*A.*) *thaliana* leaf cells treated with 100 mM NaCl [151]. In addition, plasmolysis, and irregular and broken cell walls combined with broken and obscured cell membranes, were characteristic ultrastructural features of *Brachypodium* seedling roots treated with H_2_O_2_ [66] and osmotic (15% polyethylene glycol PEG6000) stress [65] for 6 h and 2 days, respectively. Local thickening of the meristematic cell walls, which makes them wavy and heterogenous in structure, has been noted in garlic roots treated with various concentrations (80, 160, and 320 µM) of selenium (Se) for 24 h [154]. A study performed by Bilska-Kos et al. [156] demonstrated marked constriction of the cytoplasmic sleeve of the plasmodesmata at the mesophyll–bundle sheath interface in the leaves of *Miscanthus* × *giganteus* under cold (12/10 °C; day/night) conditions, which led to alterations in photoassimilate transport. Similarly, decreased photosynthetic efficiency and assimilate export from a leaf characterized by ultrastructural modifications in its plasmodesmata (strong constriction/swelling of the sphincters) crossing bundle sheath and vascular parenchyma cell/Kranz mesophyll and bundle sheath cell interfaces were also observed in a chilling-sensitive maize line chilled (14 °C) for 1 and/or 28 h [157]. 

#### 3.5.2. Nucleus

The TEM can be used to evaluate the location and quantitative relationship between condensed and decondensed chromatin, the presence and localization of nuclear bodies, and the presence of nuclear membrane invaginations and inclusions in plant cells subjected to various abiotic factors [173]. For instance, Baranova and Gulevich [158] documented an increase in the number of amoeboid nucleoli with protuberances, disturbances in chromatin compaction, and the occurrence of nuclear bodies of unknown etiology in *Triticum* (*T*.) *aestivum* apical meristem during germination under alkaline (pH 8) conditions. Treatment with H_2_O_2_ for 6 h [66] and with 15% polyethylene glycol for 2 days [65] in *Brachypodium* seedling roots resulted in damage to the nuclear membrane, and deformation and spreading of the nucleoli. In root tip cells of *Allium* (*A*.) *sativum* treated with 10^−4^ M Cu solutions for 72 h, the disruption of nuclear membranes and high condensation of chromatin material, which led to the death of some exposed cells, were observed [159]. The apical meristem cells of the root and shoot of wheat (*T*. *aestivum* L.) seedlings germinating in the presence of NaCl exhibited lumps of condensed chromatin inside the nucleus and nucleolus, indicating the inhibition of synthetic processes, including nucleolus activity. These cells exposed to Na_2_SO_4_ also exhibited significantly increased separation between condensed and decondensed chromatin. In both experiments, the appearance of nuclear invagination or complete changes in the shapes of the nuclei, as a qualitative characteristic of their damage, was also noted [152].

#### 3.5.3. Plastids (Chloroplasts and Amyloplasts)

Plastids are pivotal subcellular organelles of great importance for plants. They play a key role in many essential cellular processes, such as photosynthesis and the synthesis and storage of metabolites, and stress signaling [121]. To ensure that they can function in a broad spectrum of cell/tissue/organ-specific developmental processes and respond to a variety of environmental signals, they exist in diverse forms with different ultrastructural features and characteristics [174]. Therefore, understanding their ultrastructure and dynamics is crucial to obtain a holistic comprehension of their function in a dynamic environment.

In green leaves, the most prominent of the unique organelles are chloroplasts, consisting of thylakoids, plastoglobules, and starch, which are closely linked to their metabolism [121]. Importantly, chloroplasts have the ability to dynamically adapt their metabolic and energy-converting performance based on the needs of the plant metabolism under the given environmental conditions [175], and usually, they are the earliest abiotic damage sites visible in the plant ultrastructure [176]. In fact, ultrastructural alterations in the chloroplasts are closely interrelated with the molecular, biochemical, physiological, and functional adaptive strategies of the photosynthetic apparatus; however, they can also be associated with serious injury. When temperatures decrease, chloroplasts undergo specific ultrastructural changes, such as loss of their lenticular shape, an increased volume of stroma, along with overall increased size of the organelle (due to the accumulation of compatible osmolytes), and a decreased number and size of starch grains (due to starch hydrolyzation for the *de novo* synthesis of compatible osmoprotectants) [82]. Additionally, thylakoid membranes arranged in grana stacks can become undulated and distorted, as observed in the chloroplasts of *Arabidopsis* mesophyll cells subjected to chilling (2.5–4 °C; 14 h dark/10 h light cycle) for 72 h [160]. Similar ultrastructural changes in chloroplasts, characterized by damaged membranes and thylakoids with a loss of grana stacking, have been observed in coffee leaves (*Coffea arabica* L.) subjected to heat (37 °C) stress [155]. Swollen and deformed chloroplasts with dissolved grana thylakoid lamellae and the presence of some vesicles in the stroma were reported in mesophyll leaf cells of *Brachypodium distachyon* treated with H_2_O_2_ for 6 h [66]. These changes in chloroplast structure are indicative of their molecular, biochemical, physiological, and functional adaptations to different environmental stressors. Under high light stress conditions, across section of exposed *A*. *thaliana* leaves showed a significant reduction in chloroplast size (49%), as well as a decrease in the number of palisade cell layers (48%), spongy parenchyma (29%), and thylakoids (22%), along with a complete absence of starch. Due to the deposition of degraded thylakoid membrane components, these changes were also accompanied by a massive increase in the size of thylakoid-like lipoprotein particles, i.e., plastoglobules. The reduction in starch content appears to be an adaptive mechanism of plants experiencing high light-induced disturbed photosynthesis, as it allows for the remobilization of starch and an increase in the levels of soluble sugars to meet energy and carbon demands [121]. In salt-sensitive plant species, salinity can lead to the aggregation of chloroplasts, accompanied by the swelling of grana and fret compartments or by the complete distortion of chloroplast grana and thylakoid structures, indicating their injury [177].

In potato plantlets, an increasing concentration (25, 50, 100, and 200 mM) of NaCl and duration of treatments (2 and 6 weeks) led to a gradual reduction in the number of chloroplasts, and damage to the organelles that contained more starch. As a result, there was complete disorganization of the stroma lamellae and reduced grana stacking due to protein synthesis inhibition [153]. Similarly, Neves et al. [151] studied the effects of environmental stress on the leaf chloroplast ultrastructure of *A*. *thaliana*, and found that different stress scenarios (saline, hydric, oxidative stress, and metal poisoning) caused various changes in chloroplasts that undoubtedly affected photosynthesis. These organelles displayed dilated thylakoid membranes and the presence of several plastoglobuli inclusions and membranous structures (100 mM NaCl); a variable number of large starch grains (100 mM mannitol); destruction of the lamellar system with marked alterations in the stroma and thylakoid organization (0.5 mM H_2_O_2_); and disorganized and curved stroma and thylakoid membranes, along with the occurrence of several plastoglobuli inclusions and large starch granules (150 µM ZnSO_4_). The high amounts of carbohydrates found in these exposed chloroplasts might be due to greater intracellular compartmentalization to manage water deficit in the cells, but they might also be associated with the impairment of sugar metabolism, indicating decreased cellular respiration. A report by Baruah et al. [161] showed damaged chloroplasts with disintegrated inner membranes and disturbances in the orientation of their grana, as well as starch accumulation, in leaf cells of *Trapa (T.) natans* treated with antimony (Sb; 60 μmol/L). Swollen chloroplasts with impaired thylakoids have also been demonstrated in Fe-deficient *Cucumis sativus* [162].

In plant roots and storage tissues, amyloplasts are the most common plastids responsible for the biosynthesis and storage of starch. For instance, they are present in the central part of the root cap (RC—the columella) of young seedlings, where they also perform an essential sensory function, and in the bundle sheath that accompanies the vasculature of young shoots [178]. Inside the amyloplasts (stroma), starch is produced and formed into insoluble particles, referred to as starch grains, whose morphology is species-dependent [179]. Salinity stress is commonly associated with the rapid degradation of amyloplasts in the root columella cells of *Arabidopsis* [163] and changes in amyloplast distribution (related to their clumping around the cell nuclei) in the RC of *Pisum* (*P*.) *sativum* [164]. Baranova et al. [165] evaluated the effect of salinity (77.5 mM Na_2_SO_4_) on the ultrastructural organization of amyloplasts in the columella, peripheral zone, and initial zone of the RC of tobacco (*Nicotiana tabacum*) using TEM. Their observations showed a decrease in the number of starch grains in the amyloplasts of the columella and no amyloplasts in the peripheral zone of the RC treated with Na_2_SO_4_. In the cap initials, the researchers observed condensation of the plastid stroma and the appearance of lamellar structures and starch inclusions, which apparently have a storage function. Degradation of the starch amyloplasts in root columella cells of *Arabidopsis* and radish *(Raphanus sativus)* was also induced by a moisture gradient (water stress) [166]. This suggests that roots display hydrotropism with less interference from gravitropism. Moreover, Caldelas et al. [167] demonstrated a decrease in the size of amyloplasts in the rhizome parenchyma of *Iris pseudacorus* after exposure to chromium (CrCl_3_.6H_2_O at 0.75 mM for 5 weeks).

#### 3.5.4. Mitochondria

In general, mitochondria are highly dynamic organelles whose unique and markedly distinguished architecture is easily recognizable in electron micrographs. Along with their versatile functionality, which is reflected in the modifications in their ultrastructural features, these organelles are considered worthwhile objects of interest in basic morphological research [180,181,182]. In plant responses and stress tolerance, mitochondria play an irreplaceable role [136], and changes in their function can also be seen in their ultrastructure. In fact, several reports indicate that enhanced cellular demand for energy under stress constraints may be regulated by increasing the density of mitochondria [177] or ultrastructural rearrangements of the organelle, such as cristae morphology remodeling [162]. Additionally, it has been found that the fusion and aggregation of mitochondria can serve as a general stress response to low temperature in the palisade parenchyma cells of *Ranunculus glacialis*, in order to maintain and/or increase cellular respiration [168].

On the other hand, mitochondrial organization can be severely affected by unfavorable abiotic conditions, as has been identified in recent observations. In fact, adverse ultrastructural modifications of the organelle, reflecting its damage, have been observed in the tissues of various plant models coping with Fe-deficiency [162], high saline concentration [169], H_2_O_2_ treatment [66], osmotic stress [65], and Se toxicity [154]. This severe damage to the mitochondria included lower matrix density with a reduced number of cristae, dilated cristae, a disintegrated matrix with messy or absent cristae, and a disrupted mitochondrial outer membrane. Additionally, changes in the shape of the mitochondria (such as the formation of invaginations or even cup-shaped mitochondria) can reveal the high sensitivity of the organelle to stress, most often demonstrating disturbances in the process of division [152]. Exposure to 10^−4^ M Cu solutions gradually resulted in modifications in mitochondrial shape, due to the loss of matrix density and the extension of cisternae, in the root meristematic cells of *A*. *sativum* [159].

#### 3.5.5. Endoplasmic Reticulum, Golgi Apparatus (Dictyosomes), and Vacuoles

The ER, a central network of interconnected tubules and flattened cisternae (with the membrane accounting for approximately 50% of total cellular membranes), is a main stress sensor in plant cells. Indeed, it serves as the starting point for stress responses, and from here, proteins and signals are either distributed to other cellular locations or degraded [183,184]. For sorting proteins to their final destinations, such as vacuoles, macromolecules are first exported from the ER to the GA, which are central components of the endomembrane trafficking system [185,186]. Additionally, the ER plays a vital role in maintaining the homeostasis of non-secretory organelles such as mitochondria and chloroplasts [185]. A study by Neves et al. [151] has shown that abiotic stress greatly affects endomembrane trafficking pathways, with changes beginning at the subcellular level that involve rearrangements of the endomembrane system. In *A. thaliana* grown under oxidative stress conditions (0.5 mM H_2_O_2_), hypertrophied GA, particularly evident at the swollen edges of the cisternae in root cells, were noted in both leaf and root segments. Moreover, many membrane formations in the cytoplasm or associated with the vacuoles were observed in treated leaf cells, indicating a high degree of membrane remodeling, as shown by TEM. Glińska et al. [154] reported that after treating meristematic cells of garlic (*A*. *sativum* L.) roots with Se (80, 160, and 320 µM) for 24 h, the cells displayed abundant ER cisternae arranged in concentric or parallel orientation to the cell wall, along with significant ultrastructural changes in the GA. In these cells, the reduced number of cisternae per dictyosome, together with the increased number of vesicles (originating from dictyosomes) or even deformed Golgi cisternae with numerous membrane invaginations, were also frequently observed. Additionally, vacuoles varying in size and shape, filled with wall-like material, were present near the nucleus. In root samples of garden pea (*P*. *sativum*) exposed to tungsten (W; 200 mg/L) for 24 h, ribosome-bearing cisternae of the ER with concentric conformations frequently enclosed cytoplasmic organelles (such as plastids or mitochondria), and Golgi bodies were degraded and had no surrounding vesicles, indicating inactivity. Additionally, collapsed and deformed vacuoles were present in the samples [170]. In garlic root tip cells, Cu (10^−4^ M) treatment following varying exposure durations induced gradual ultrastructural modifications. These included a significant increase in dictyosome vesicles (after 1 h), dilation of the flattened cisternae of the ER (after 2 h) and their disintegration into small closed vesicles with electron-dense granules (after 4 h), and the formation of larger vacuoles with more electron-dense granules (after 8 h). Interestingly, after 12 h of Cu treatment, the cells exhibited an abundance of parallel arrays of rough ER with regular cisternae [159]. Finally, broken ER scattered close to the plasma membrane was observed in *Brachypodium* seedling roots exposed to H_2_O_2_ treatment (for 6 h) [66]. 

Vacuoles are highly dynamic and pleiomorphic organelles that occupy a large volume (up to 90%) of plant cells and perform multiple functions. They change their size under varying growth conditions [187]. These organelles play a fundamental role in defensive plant mechanisms against abiotic stress by accumulating toxic products and maintaining cell turgor pressure [184]. Through their ability to regulate the exchange of fluids and ions between the cytoplasm and the intravacuolar solution, vacuoles can rapidly reduce osmotic pressure (turgor) under osmotic stress conditions [152,187]. In the cotyledons and seedling roots of *Medicago sativa* L., depending on the osmotic effect of salinity, modifications in the quantity and form of residual protein bodies within vacuoles, and in the conversion of storage vacuoles to central ones, can be observed using TEM [171]. Moreover, as one of the main storage sites of heavy metals, vacuolar compartmentalization is an indispensable component of heavy metal detoxification in plants [172], as shown in the above-mentioned studies. In research conducted by Baruah et al. [161], the subcellular compartmentalization of Sb (in the form of an electron-dense precipitate) was found in the vacuoles of root cells of *T. natans*, which prevented its high concentration in the cytosol. Thus, vacuoles play a crucial role in the detoxification and storage of harmful substances in plant cells.

Moreover, obtaining both qualitative and quantitative data on subcellular compartments is crucial when studying the relationship between ultrastructure and function in plants experiencing abiotic stress factors (Table 3). This approach enables a more accurate interpretation of results obtained from biochemical analysis, which sheds more light on plants’ adaptive capacity. These helpful quantitative estimates of subcellular compartments by means of TEM have been utilized in numerous studies [160,188,189,190,191,192,193,194] to achieve this goal.

## 4. Root Cap as a Valuable Model System for Study of Ultrastructural Plant Responses to Abiotic Stress 

Roots have a plethora of primary functions, including anchoring the plant in the soil, absorbing and conducting water and dissolved nutrients and minerals, and storing reserve substances. Therefore, they are critical organs for the productivity and adaptation of plants [195]. Indeed, root development exhibits high flexibility and plasticity, allowing for plant acclimation and survival under abiotic stress by adjusting root morphology in response to environmental changes [196,197]. In this sense, the root tip has the ability to act as a finely tuned sensor for various stress circumstances [177]. The apical part of the roots (root tips) of most higher plants (including almost all crop species) is covered with a multilayered dome of spindle-shaped parenchyma cells, known as RC or calyptra [165,198]. Owing to its advance-guard position, the RC not only provides protective functions to the delicate stem cells in the root meristem within the root tip, but also receives and transmits environmental signals (on factors such as pressure, moisture, and gravity) to the inner root tissues [199]. In addition, the last layer of the RC, known as root border cells (BCs), is mixed with RC mucilage, which effectively creates favorable conditions for the existence of plant growth-promoting bacteria in the rhizoplane and rhizosphere zones. This complex also reduces the resistance roots experience during penetration into field soil. In fact, the presence of RC alone can reduce about 30–40% of the resistance [165,198]. Border cells released from the root tip in the form of an organized layer of several remaining attached cells are termed “border-like cells” (BLCs), which also protect the root from both biotic and abiotic stresses [200,201]. 

The RC also plays an important role in gravity perception. To perform all of its functions, the position and size of the RC remain stable throughout root growth, thanks to the rapid and constant turnover of short-lived cells (the replacement of old BLCs with new ones). This turnover is regulated by an intricate balance of cell generation, differentiation, and degeneration [199,201]. In maize, the life cycle of RC cells is estimated to range from one to seven days, with up to 3200 sloughed BLCs (approximately 4000 to 21,000 presented in a complete maize RC) found in rhizosphere soil per root per day [198]. Given these characteristics, the RC is an ideal model system for studying plant developmental changes under different abiotic conditions, both spatially and temporally. 

Structurally, the RC creates a small dome of cells surrounding the root apical meristem (RAM), which is a tissue composed of actively dividing cells that are responsible for producing all cells in the root system [201]. The RAM is composed of two groups of undifferentiated cells: (i) the mitotically less-active quiescent center (QC) and (ii) the highly mitotically active initials surrounding the QC [202]. In this sense, the RC originates from the calyptrogen (RC cell-initials), which is the distal part of the RAM [165,202,203,204]. Recently, Berhin et al. [205] discovered that the first layer of the primary root RC is covered by a hitherto unrecognized electron-opaque cell wall modification resembling a plant cuticle, known as the RC cuticle (RCC). Moreover, their findings revealed that the RCC acts as a diffusion barrier that protects the vulnerable seedling meristem under osmotic stress and high salt conditions, thus supporting seedling survival and adaptation to environmental challenges. The RC itself is strictly organized into two zones: (i) the columella (occupying its central part) and (ii) the peripheral zone of the RC, i.e., the lateral RC (covering the columella). The cells of the peripheral zone are highly vacuolated and have a smaller cytoplasm volume. They extend to the apical meristem cells and the distal part of the root extension zone [165]. Histologically, the columella is composed of two different cell types: (i) the statocytes, mostly located in three to four internal layers, sometimes referred to as tiers or stories (S1, S2, S3), of the columella, and (ii) the secretory cells constituting the lateral RC and the external columella layer [202,206,207]. Statocytes contain amyloplasts whose starch inclusions act as statoliths that sediment in the direction of the gravity vector in the root tips. So, the gravity sensing structures control and determine the growth angle of the plant roots [197,198]. In addition to these small starch grains (which are almost not utilized in the plant metabolic process), amyloplasts also store ordinary starch granules with a different structure. In fact, statolith amyloplasts possess a 0.1–0.3 µm thick boundary layer surrounding the starch granules, linking them into a single structural unit within each amyloplast [203]. Due to the accumulation of starch inclusions, amyloplasts significantly increase the size of not only the plastids but also the cells themselves. Moreover, because of the presence of numerous vacuoles in the cytoplasm, statocytes are commonly larger in size compared to meristem cells [165].

### Ultrastructure of Statocytes in Plant Root Tips

Typically, the central columella cells of RC (Figure 4) found in diverse plant species exhibit a distinct structural polarity, which is thought to be connected to their gravity-sensing ability. This polarity is characterized by the presence of a peripheral network of tubular ER located beneath the plasma membrane, nuclei positioned in the central or proximal region of the cells, and amyloplasts containing starch granules (Figure 3C–E) that serve as statoliths, situated in the distal portion of the statocytes. Additionally, several separate, enlarged vacuoles (Figure 3B), and mitochondria (Figure 3B,F), are dispersed throughout the cytoplasm. The cisternae of GA can also be observed in columella cells [165,207,208,209]. In the central region of statocytes, the cytosol occupies an unusually large proportion of the cytoplasmic volume compared to other root tip cells, and is devoid of actin filament bundles [203]. Microtubules are only located at the cell periphery [208]. On the contrary, the cytosol appears to consist of randomly oriented single actin filaments forming (together with other molecules) a network-like cytoskeletal matrix [203]. The structural asymmetry of the cell is maintained with microtubules and the actin cytoskeleton [210], and it is accompanied by the initial sedimentation of amyloplasts during the differentiation of meristematic calyptrogen cells into columella ones [211]. This is because of the higher density of amyloplasts (1.5 g/cm^3^) compared to the surrounding cytoplasm (1.02–1.1 g/cm^3^) [165,212]. So, the statolith sedimentation rate depends on their number, size, and density, as well as the strength of their interaction with actin filaments [210]. In fact, amyloplasts are finely suspended by actin filaments (microfibrils) on a bed of asymmetrically disposed ER membranes anchored by cytoskeletal elements (including microtubules) to the plasmalemma in the lower part of the cells. In gravistimulated roots, starch-filled plastids are displaced to a new bed of ER membranes, and this displacement (perceived by the ER and plasma membranes) consequently activates a signal transduction pathway, resulting in a physiological signal responsible for root curvature at the response site [209]. In the absence of gravity, amyloplasts have the ability to perform myosin-mediated autonomous movement. Moreover, the polarity of the statocytes is also evident in the distribution of plasmodesmata, which are more numerous in the transverse walls than in the longitudinal ones [208]. Regarding the ER, Zheng and Staehelin [211] observed a specialized form of ER, termed the “nodal ER”, in the statocytes of tobacco root tips preserved using high-pressure freezing/freeze-substitution techniques. The unique domain is characterized by the presence of a central “nodal rod” element to which are attached 3–8 (usually 7) rough ER cisternae. The edge-on attachment of the ER membranes to the nodal rod appears to be partly responsible for the sheet-like organization of the ER membranes, an organization that stimulates the binding of polysomes [203]. The nodal rods, which have a diameter of approximately 100 nm, are composed of more lightly staining, oblong subunits with a diameter and length of approximately 10 nm and 20 nm, respectively [211].

Importantly, with their unique ultrastructural characteristics, the architecture of statocytes can be effectively employed as a basis for assessing the impacts of various abiotic factors related to plant stress responses. These gravity-sensing cells have been widely used as a model system, especially for studying plant growth under a variety of gravity conditions [207,213,214,215,216,217,218]. However, there are also numerous reports dealing with ultrastructural modifications in the columella statocytes induced by diverse chemical substances, such as carboxylic acid antibiotic A23187 (calcimycin) [219], Ca^2+^ channel blockers (D600 and nicardipine) [220], lithium (LiCl) [221] and lanthanum (LaCl_3_) [222] ions, and herbicide fluazifop-p-butyl (FPB) [223], or high-gradient magnetic fields [224]. In these studies, disruption [222] or loss of cellular polarity [220,221], pronounced vacuolization [219,220,221,222], modifications in dictyosome structure [219,220], amyloplast distribution around the nucleus [219] and in a central part of the cells [222], considerable cellular lengthening, ER fragmentation [219], thinner cell walls [219,222], amyloplast cluster appearance, condensed mitochondria, and local dilations in a perinuclear [221] and periplasmic space [222] in pea root statocytes have been observed. Abnormalities in cell shape and cell walls, along with increased vacuolization, nuclear membrane degeneration, a decreased number of subcellular organelles (mainly mitochondria), and smaller starch grains in amyloplasts, were reported in the columella cells of lentil roots after treatment with FPB in a dose-dependent manner [223]. By contrast, a high-gradient magnetic field had no demonstrable impact on the volume of individual pea statocytes or their amyloplasts, the relative volumes of cellular organelles (except vacuoles), the number of amyloplasts per statocyte, or the surface area of the ER [224].

## 5. Conclusions

Owing to their sessile lifestyles, plants have developed a myriad of adaptive pathways and strategies to deal with stressful environments. Drought, extreme temperature, high salinity, metal and nanoparticle toxicity, and UV irradiation represent some of the most common environmental stress factors that negatively affect plant growth and development, leading to a dramatic decline in worldwide plant productivity. Abiotic stress targets biochemical pathways and affects crosstalk among the signaling pathways, which, in turn, results in the disruption of their physiological functions. Although genetic, biochemical, and molecular studies on crops or model plants have largely improved our understanding of abiotic stress responses at multiple molecular levels, much still remains undisclosed. It is well-established that elevated ROS production is the primary cause of the negative impact of abiotic stresses, as it damages macromolecules such as lipids, DNA, and proteins. The vast domain of plant stress responses includes the antioxidant activity of molecules and compounds of enzymatic or non-enzymatic origin, with the assistance of osmoregulatory and protein quality control systems and UFAs as general defenders. These diverse physiological and molecular defense approaches are also usually reflected at the subcellular level. One promising technique for evaluating plant responses to various stresses is, thus, the evaluation of ultrastructural changes using TEM. Hence, ultrastructural modifications of organelles, especially those related to plastid (mainly chloroplasts), mitochondrial, membrane, and endomembrane system reorganization (the most common ones), can represent a good indicator of plant sensitivity and/or adaptive responses to various stress scenarios. Generally, columella cells of the RC are highly organized and dynamic structures that are considered to comprise the gravity sensing site. Along with specific ultrastructure, statocytes appear to be an appropriate model system to study the actions of various abiotic stress factors, not only those related to gravitropism. Understanding these responses has significant implications for the development of engineered crops for modern agriculture.

## Figures and Tables

**Figure 1 plants-12-01666-f001:**
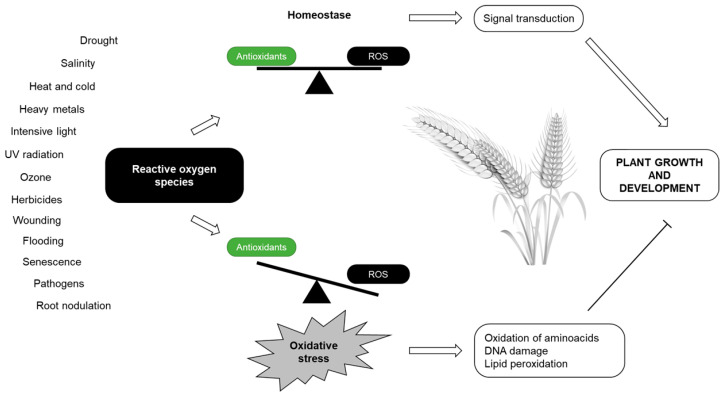
Environmental conditions that induce activation of stress response pathways while not disturbing the redox balance of the cell are capable of inducing cell adaptation that promotes growth and development. However, stress that alters the redox balance and impairs cell homeostasis leads to oxidative damage to cellular compartments and ultimately results in cell or organism death.

**Figure 2 plants-12-01666-f002:**
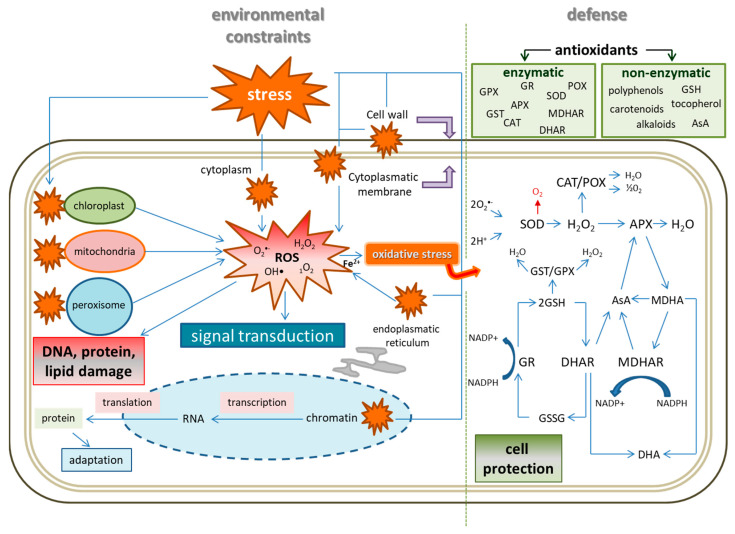
Stress acting through various cellular compartments triggers signaling events that involve second-wave messengers and regulatory proteins. Stress-derived changes in gene transcription, RNA processing, or post-translational modifications of formed proteins play crucial roles in the production and modification of proteins, triggering various responses to stress ranging from cell adaptation to apoptosis. Moreover, ROS signal sensors/receptors may induce activation of biosynthesis/functioning of different protective molecules (enzymatic or non-enzymatic antioxidants) capable of modulating individual metabolic and physiological pathways, thereby contributing to plant acclimation to various stresses (adapted from Zhang et al. [17] and Kapoor et al. [150]).

**Figure 3 plants-12-01666-f003:**
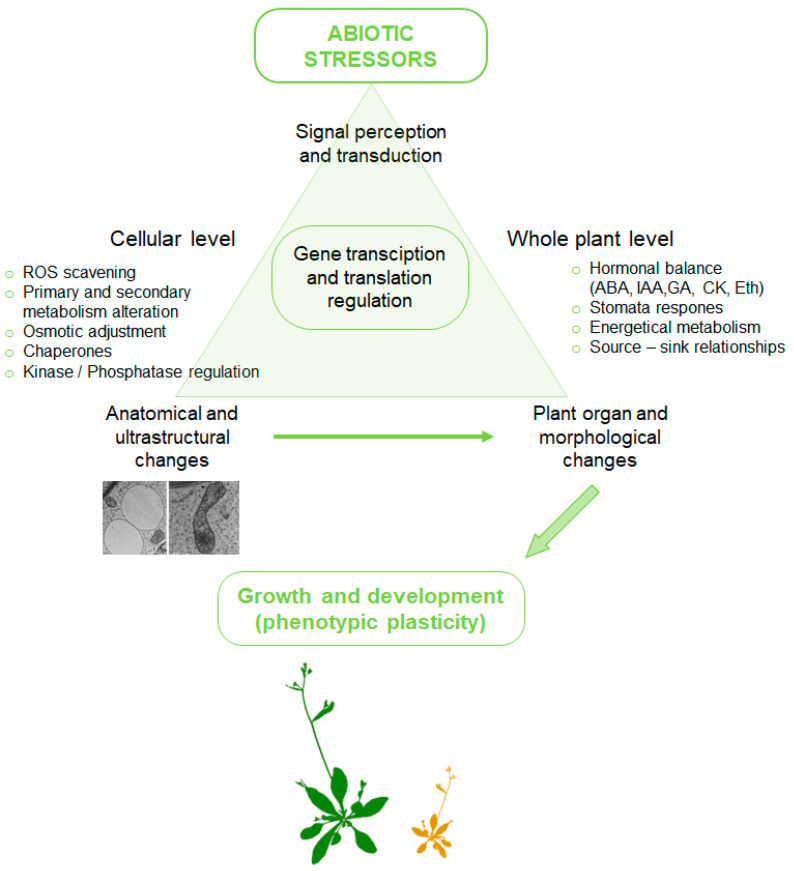
Hierarchic concept of plant phenotype modifications under abiotic stress. Stress caused by abiotic environmental factors induces, on one hand, whole-plant changes that result in plant organ and morphology abnormalities, and on the other hand, plant alterations at the cellular level resulting in anatomic and ultrastructural modifications. These changes arise from signal transduction discrepancies and/or impairments in the regulation of gene expression, and often result in growth and development retardation or death of the plant organism.

**Figure 4 plants-12-01666-f004:**
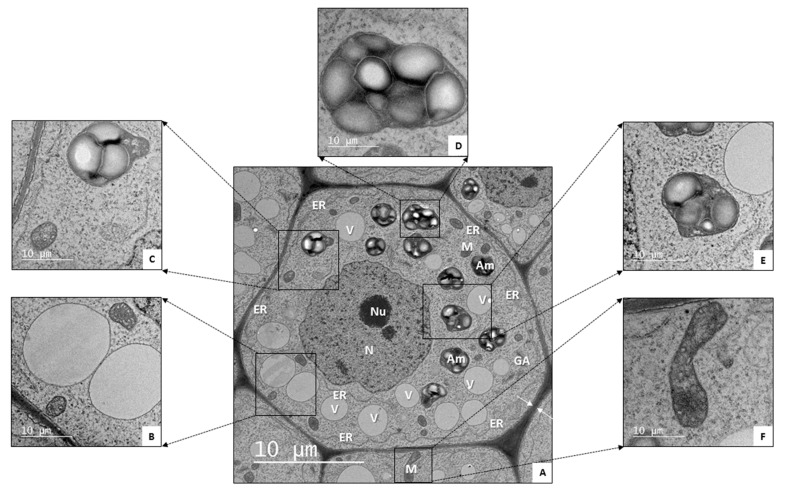
Central columella cells in the root tip of *Zea mays*. Micrographs demonstrating architecture of the RC statocytes (**A**) by means of transmission electron microscopy (TEM; JEM-2100 JEOL, Tokyo, Japan, operating at 200 kV), and a more detailed ultrastructural organization of the tubular type of mitochondria (**B**,**F**), and starch-rich plastids (amyloplasts) (**C–E**). Note: N—nucleus, Nu—nucleolus, V—vacuole, ER—endoplasmic reticulum, Am—amyloplast, M—mitochondrion, GA—Golgi apparatus, facing arrows—cell wall; scale bars: 10 µm (magnification ×800 (**A**); ×4000 (**B**,**C**,**E**); ×6000 (**D**,**F**)).

**Table 1 plants-12-01666-t001:** Electron donation sources causing ROS generation.

Compound	Notations	Sources
Singlet oxygen	^1^O_2_	UV radiation, photoinhibition, photosystem II electron transfer reactions
Superoxide anion	O_2_•^−^	Mehler reaction in photosynthetic electron transport, photorespiration, mitochondrial electron transport, plasmalemma, nitrogen fixation, O_3_ and OH^−^ reactions in apoplast, pathogen defense responses
Hydrogen peroxide	H_2_O_2_	Photorespiration, β-oxidation, pathogen defense responses
Hydroxyl radical	OH•	O_3_ reaction in apoplast, pathogens defense responses
Perhydroxyl radical	O_2_H•	O_3_ and OH^−^ reactions in apoplast

**Table 2 plants-12-01666-t002:** Effects of various abiotic stress factors on qualitative ultrastructural characteristics of plant model systems.

Compartment/Structure	Abiotic Factor	Plants Species	Cell Effects	Reference
Cell wall	Salinity (25–200 mM NaCl)	*Solanum tuberosum*	Twisted and ruptured cell walls, cell wall disintegration	[153]
	Salinity (100 mM NaCl)	*Arabidopsis thaliana*	Detachment of plasmalemma from cell wall	[151]
	Oxidative stress (20 mM H_2_O_2_)	*Brachypodium distachyon*	Broken cell walls	[66]
	Osmotic stress (15% PEG6000)	*Brachypodium distachyon*	Broken cell walls	[65]
	Metal stress (80–320 μM Na_2_SeO_4_)	*Allium sativum*	Local thickenings of cell walls	[154]
	Heat stress (37 °C)	*Coffea arabica*	Modification of cell wall structure	[155]
Plasma membrane	Salinity (25–200 mM NaCl)	*Solanum tuberosum*	Membrane invagination	[153]
	Osmotic stress (15% PEG6000)	*Brachypodium distachyon*	Obscured cell membranes, plasmolysis	[65]
Plasmodesmata	Cold stress (12/10 °C day/night)	*Miscanthus × giganteus*	Marked constriction of the cytoplasmic sleeve of the plasmodesmata at the mesophyll–bundle sheath interface	[156]
	Chilling stress (14 °C)	*Zea mays*	Strong constriction and swelling of the sphincters in plasmodesmata	[157]
Nucleus	Alkaline stress (pH 8.0)	*Triticum aestivum*	Increase in the number of amoeboid nucleoli with protuberances, disturbances in chromatin compaction, and occurrence of nuclear bodies of unknown etiology	[158]
	Oxidative stress (20 mM H_2_O_2_)	*Brachypodium distachyon*	Damage to nuclear membrane	[66]
	Osmotic stress (15% PEG6000)	*Brachypodium distachyon*	Deformation and spreading of nucleoli	[65]
	Metal stress (0.1 mM CuSO_4_)	*Allium sativum*	Disruption of nuclear membranes and high condensation of chromatin	[159]
	Salinity (0.1 M NaCl and 0.1 M Na_2_SO_4_)	*Triticum aestivum*	Lumps of condensed chromatin inside the nucleus and nucleolus, increased separation between condensed and decondensed chromatin, appearance of nucleus invagination, and complete change in the shape of the nuclei	[152]
Plastid	High light stress (1500 μmol m^−2^ s^−1^) Pathogen infection (*Botrytis cinerea*) Dark-induced senescence	*Arabidopsis thaliana*	Significant decrease in chloroplast number, chloroplast size reduction, thylakoid area reduction, increase in plastoglobules, changes in starch content	[121]
	Chilling stress (2.5–4 °C)	*Arabidopsis thaliana*	Undulated and distorted thylakoid membranes arranged in grana stacks, large accumulation of starch, increase in average area per chloroplast	[160]
	Heat stress (37 °C)	*Coffea arabica*	Changes in thylakoid integration, loss of grana stacking	[155]
	Oxidative stress (20 mM H_2_O_2_)	*Brachypodium distachyon*	Swollen and deformed chloroplasts with dissolved grana thylakoid lamellae	[66]
	Salinity (25–200 mM NaCl)	*Solanum tuberosum*	Aggregation of chloroplasts accompanied by swelling of grana and fret compartments or by the complete distortion of chloroplast grana and thylakoid structures	[153]
	Salinity (100 mM NaCl)	*Arabidopsis thaliana*	Dilated thylakoid membranes, plastoglobuli accumulation	[151]
	Osmotic stress (100 mM mannitol)	*Arabidopsis thaliana*	Large starch grain accumulation	[151]
	Oxidative stress (0.5 mM H_2_O_2_)	*Arabidopsis thaliana*	Destruction of the lamellar system, marked alterations in stroma and thylakoid organization	[151]
	Metal stress (150 μM ZnSO_4_)	*Arabidopsis thaliana*	Disorganized and curved stroma and thylakoid membranes, occurrence of several plastoglobuli inclusions and large starch granules	[151]
Plastid	Metal stress (100 μM SbCl_3_)	*Trapa natans*	Damaged chloroplasts, disintegrated inner membrane, disturbances in the orientation of the grana, starch accumulation	[161]
	Metal stress (0.1 mM Fe(III)-EDTA)	*Cucumis sativus*	Swollen chloroplasts and impaired thylakoids	[162]
Amyloplast	Salinity (25–125 mM NaCl)	*Arabidopsis thaliana*	Rapid degradation of amyloplasts	[163]
	Salinity (120 mM NaCl)	*Pisum sativum*	Changes in amyloplast distribution	[164]
	Salinity (77.5 mM Na_2_SO_4_)	*Nicotiana tabacum*	Decrease in the number of starch grains in amyloplasts of the columella, no amyloplasts in the peripheral zone of the root cap	[165]
	Osmotic stress (sorbitol solution)	*Arabidopsis thaliana and Raphanus sativus*	Degradation of the starch amyloplasts in root columella cells	[166]
	Metal stress (0.75 mM CrCl_3_)	*Iris pseudacorus*	Decrease in the size of amyloplasts in the rhizome parenchyma	[167]
Mitochondria	Metal stress (0.1 mM Fe(III)-EDTA)	*Cucumis sativus*	Mitochondria rearrangements, cristae remodeling	[162]
	Chilling stress (0.5–4 °C)	*Ranunculus glacialis*	Fusion and aggregation of mitochondria	[168]
	Salt stress (86.2–258 mM NaCl) Oxidative stress (20 mM H_2_O_2_) Osmotic stress (15% PEG6000) Metal stress (80–320 μM Na_2_SeO_4_)	*Nicotiana tabacum* *Brachypodium distachyon* *Allium sativum*	Lower matrix density with reduced number of cristae, dilated cristae, disintegrated matrix with messy or absent cristae	[65,66,154,169]
	Alkaline stress (pH 8.0)	*Triticum aestivum*	Formation of invaginations or even cup-shaped mitochondria	[152]
	Metal stress (0.1 mM CuSO_4_)	*Allium sativum*	Modifications in mitochondrial shape in the root meristematic cells, loss of matrix density, and an extension of cisternae	[159]
Mitochondria	Alkaline stress (pH 8.0)	*Triticum aestivum*	Formation of invaginations or even cup-shaped mitochondria	[152]
	Metal stress (0.1 mM CuSO_4_)	*Allium sativum*	Modifications in mitochondrial shape in the root meristematic cells, loss of matrix density, and an extension of cisternae	[159]
Endoplasmic reticulum	Salinity (100 mM NaCl)	*Arabidopsis thaliana*	Endomembrane rearrangements	[151]
	Metal stress (80–320 μM Na_2_SeO_4_)	*Allium sativum*	Appearance of concentric or parallel arrangement of abundant ER cisternae	[154]
	Metal stress (200–500 mg/L Na_2_WO_4_)	*Pisum sativum*	Ribosome-bearing cisternae of ER with concentric conformations frequently enclose cytoplasmic organelles	[170]
	Metal stress (0.1 mM CuSO_4_)	*Allium sativum*	Dilation of flattened cisternae of ER and their disintegration into small closed vesicles	[159]
	Oxidative stress (20 mM H_2_O_2_)	*Brachypodium distachyon*	Broken ER scattered close to the plasma membrane	[66]
Golgi apparatus	Oxidative stress (0.5 mM H_2_O_2_)	*Arabidopsis thaliana*	Hypertrophied Golgi, high degree of membrane remodeling	[151]
	Metal stress (80–320 μM Na_2_SeO_4_)	*Allium sativum*	Significant ultrastructural changes in GA	[154]
	Metal stress (0.1 mM CuSO_4_)	*Allium sativum*	Increase in dictyosome vesicles	[159]
Vacuole	Metal stress (200–500 mg/L Na_2_WO_4_)	*Pisum sativum*	Deformation and variation in size and shape of vacuoles	[170]
	Metal stress (0.1 mM CuSO_4_)	*Allium sativum*	Formation of larger vacuoles	[159]
	Salinity (NaCl, Na_2_SO_4_) and Osmotic stress (mannitol),	*Medicago sativa*	Modifications in the quantity and form of residual protein bodies within vacuoles	[171]
	Metal stress (100 μM SbCl_3_)	*Trapa natans*	Accumulation of Sb in vacuoles	[161]
	Metal stress (20–60 μμM CdCl_2_)	*Oryza sativa*	Cd compartmentation in vacuoles	[172]

**Table 3 plants-12-01666-t003:** Effects of various abiotic stress factors on quantitative ultrastructural characteristics of plant model systems.

Compartment/Plant Organ	Abiotic Factor	Plant Species	Cell Effects	Reference
Leaf—mesophyll cells	Chilling (2.5–4 °C in the dark and 3.2–4 °C in the light for 72 h)	*Arabidopsis thaliana* (Col 0)	Increased average area per chloroplast in cell sections	[160]
			Reduced chloroplast size	
			Significantly higher abundance of ring-shaped and other morphologically aberrant mitochondria	
Leaf—mesophyll cells	Drought (induced by slowly decreasing the amount of supplied water over a time period of 4 weeks)	*Spinacia oleracea* L. cv. Matador	Increased absolute total volume and surface area of chloroplasts	[188]
			Increased volume of stroma and thylakoids, and increased thylakoid surface area in chloroplasts	
			Lack of starch grains	
			Decreased mean volume and surface area of mitochondria	
Leaf—mesophyll cells	Chilling (18 °C during day and 8 °C during night for four weeks)	*Zea mays* L.	Suppressed development of the system of thylakoids, and decreased volume and surface density of all thylakoids in the chloroplasts	[189]
			Decreased volume and surface density of intergranal thylakoids in chloroplasts	
Leaf—mesophyll cells	Drought (induced by withholding watering for 7 days)	*Triticum aestivum* L.	Increased proportion of spherical and oval-shaped mitochondria	[190]
			Increased mean size of mitochondria	
			Decreased relative cell area occupied by mitochondria in the drought-sensitive varieties	
Leaf—mesophyll cells	Drought (induced by withholding watering for 7 days)	*Triticum aestivum* L. (drought-sensitive variety)	Increased size of chloroplasts, mitochondria, and plastoglobules	[191]
			Increased number of chloroplasts and plastoglobules per 100 µm^2^ visible field	
			Decreased number of mitochondria per 100 µm^2^ visible field	
	High temperature (40 °C for 5 h)		Increased size of chloroplasts, mitochondria, and plastoglobules	
			Increased number of chloroplasts and mitochondria per 100 µm^2^ visible field	
	Drought + high temperature		Increased size of chloroplasts, mitochondria, and plastoglobules	
			Increased number of chloroplasts, mitochondria, and plastoglobules per 100 µm^2^ visible field	
Apical layer of curds	Cold stress (8 °C for 10 days) and heat stress (40 °C for 4 h)	*Brassica oleracea* var. *botrytis*	No significant changes in mitochondrial number and mitochondrial area per field	[192]
			Decreased mitochondrial number per field after stress (heat and cold) recovery	
Leaf—mesophyll cells	Drought (simulated by 20% polyethylene glycol 6000 (−0.6 MPa) for 2 days)	*Zea mays* L.	No significant changes in the cell area occupied by chloroplasts and in the size and length-to-width ratio of chloroplasts in the drought-resistant line	[193]
			Significant reduction in the length-to-width ratios of chloroplasts and the cell area occupied by chloroplasts in drought-sensitive lines	
Needle—mesophyll cells	Air CO_2_	*Picea abies* L. Karst.	Increased number of chloroplasts per mesophyll volume (sampled systematically, uniformly, and randomly from the whole needle cross-section area)	[194]
			Increased starch areal density and starch grain area in chloroplasts (sampled from both the whole needle cross-section area and from the first layer of mesophyll)	
	Irradiance		Increased starch areal density and starch grain area in chloroplasts (sampled from both the whole needle cross-section area and from the first layer of mesophyll)	

## Data Availability

Not applicable.

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
