# Peer review of "Modifications in Ultrastructural Characteristics and Redox Status of Plants under Environmental Stress: A Review"

_plants, 2023, doi:10.3390/plants12081666_

Round 1

Reviewer 1 Report

The aim of paper was to show how abiotic factors affecting plants and their response to stress, with a particular focus on the impact of oxidative changes and alterations in cellular ultrastructure.The authors have done a good job, they present a solid review based on a large number of works (224). 

In terms of content, I have no serious comments, however, in the article the authors extensively describe the importance of ultrastructural studies in the study of stress, while they present only one table with such data. Therefore, I encourage the authors to add at least two more tables with ultrastructural data.

Author Response

The authors thank Reviewer 1 for the positive comments and for the suggestion.

Two tables, Table 2 and Table3, have been added describing the effects of various abiotic stress factors on qualitative and quantitative ultrastructural characteristics of plant model systems

In the line 536 a note has been included referring the Table 2.

In the line 782 a note has been included referring the Table 3.

Reviewer 2 Report

This manuscript covers a very wide range of topics. While helpful as an overview it lacks depth in exploring the topics. In some ways it seems more like a textbook chapter rather than a helpful review. Some suggestions:

1. Utilize more illustrations.

2. The title is too long and confusing. Please consider shortening and making it more clear.

3. The entirety of the text needs to be edited for more clear English. There are some typos (line 281, preocess should be process) and awkward wording (line 125, saline soils instead of salinity soils; line 433, crop quality and quantity should be crop quality and yield).

4. Line 423, why don't you write 800 and 75 Ma instead of 0.8 Ga and 75 Ma?

Author Response

The authors thank the reviewer for the comments and suggestions

  1. Utilize more illustrations.

New figure has been added

  1. The title is too long and confusing. Please consider shortening and making it more clear.

The title has been shoortened.

  1. The entirety of the text needs to be edited for more clear English. There are some typos (line 281, preocess should be process) and awkward wording (line 125, saline soils instead of salinity soils; line 433, crop quality and quantity should be crop quality and yield).

We thank the reviewer for the suggestion, the entire text has been edited for more clear English.

  1. Line 423, why don't you write 800 and 75 Ma instead of 0.8 Ga and 75 Ma?

We thank the reviewer for this point, it has been corrected.

Reviewer 3 Report

The review represents a useful and well documented upgrading concerning main abiotic stressors which highly negatively affect crop yield and related food supply. Each topic is well organised and developed and the part concerning “subcellular organelles as abiotic stress marker and its adaptive strategies” is highly relevant.

Only a small suggestion: add 1 or 2 lines on upregulation of antioxidant enzyme activities in the paragraph “Salinity (salt stress)” in  the part (lines 138 – 146) concerning mechanisms adopted by the plants to cope with salinity

Author Response

The authors thank Reviewer 2 for the positive comments and for the suggestion.

Between lines 131-135 sentences have been added to emphasize the antioxidant enzymes activities upon salinity stress in plants.

Round 2

Reviewer 2 Report

Thank you for addressing the comments and clarifying the title and some areas of the text.